# The Association between ADHD and Environmental Chemicals—A Scoping Review

**DOI:** 10.3390/ijerph19052849

**Published:** 2022-03-01

**Authors:** Sonja Moore, Laura Paalanen, Lisa Melymuk, Andromachi Katsonouri, Marike Kolossa-Gehring, Hanna Tolonen

**Affiliations:** 1Department of Public Health and Welfare, Finnish Institute for Health and Welfare (THL), 00271 Helsinki, Finland; smoore@student.uef.fi (S.M.); hanna.tolonen@thl.fi (H.T.); 2Institute of Public Health and Clinical Nutrition, Kuopio Campus, University of Eastern Finland (UEF), 70210 Kuopio, Finland; 3RECETOX, Faculty of Science, Masaryk University, 62500 Brno, Czech Republic; melymuk@recetox.muni.cz; 4State General Laboratory, Ministry of Health, Nicosia 2081, Cyprus; akatsonouri@sgl.moh.gov.cy; 5German Environment Agency (UBA), 06844 Dessau-Rosslau, Germany; marike.kolossa@uba.de

**Keywords:** ADHD, chemical exposure, HBM4EU, health effect, lead, phthalates, bisphenol A, polycyclic aromatic hydrocarbons, mercury, pesticides

## Abstract

The role of environmental chemicals in the etiology of attention deficit hyperactivity disorder (ADHD) has been of interest in recent research. This scoping review aims to summarize known or possible associations between ADHD and environmental exposures to substances selected as priority chemicals of the European Human Biomonitoring Initiative (HBM4EU). Literature searches were performed in PubMed to identify relevant publications. Only meta-analyses and review articles were included, as they provide more extensive evidence compared to individual studies. The collected evidence indicated that lead (Pb), phthalates and bisphenol A (BPA) are moderately to highly associated with ADHD. Limited evidence exists for an association between ADHD and polycyclic aromatic hydrocarbons (PAHs), flame retardants, mercury (Hg), and pesticides. The evidence of association between ADHD and cadmium (Cd) and per- and polyfluoroalkyl substances (PFASs) based on the identified reviews was low but justified further research. The methods of the individual studies included in the reviews and meta-analyses covered in the current paper varied considerably. Making precise conclusions in terms of the strength of evidence on association between certain chemicals and ADHD was not straightforward. More research is needed for stronger evidence of associations or the lack of an association between specific chemical exposures and ADHD.

## 1. Introduction

Attention deficit hyperactivity disorder (ADHD) is a chronic neurodevelopmental disorder characterized by persistent inattention and/or hyperactivity-impulsivity, which has a direct negative impact on academic, occupational, or social functioning [1]. The onset of symptoms occurs in early to mid-childhood, must be present in more than one setting (e.g., school, home), and must last for more than 6 months.

Systematic reviews have estimated the worldwide prevalence of ADHD to be between 2% and 7%, averaging around 5% in children and adolescents [2], and despite concerns of increased rates of diagnosis, the prevalence has remained steady over the past decades [3]. ADHD is more commonly diagnosed in males than females (ratio 2–3:1) but there is evidence to suggest that ADHD is under-recognized in females [2,4]. ADHD is diagnosed clinically by evaluating current and previous symptoms and functional impairment, usually with the help of parent and teacher input [5]. Several diagnostic tools, such as the Vanderbilt ADHD Diagnostic Teacher Rating Scale, the Vanderbilt ADHD Diagnostic Parent Rating Scale, Conners Rating Scale, and Diagnostic Interview for Children and Adolescents (DICA-IV) exist to aid diagnosis [6].

Children with ADHD have an increased risk of accidental injuries, worse quality of life, earlier use of tobacco and drugs, and earlier sexual activity. Although ADHD does not directly cause mortality, there is increased mortality among people with ADHD by external and accidental causes [7]. ADHD is often associated with comorbidities, such as autism spectrum disorder, intellectual disability, disruptive disorders, tic disorders, mood disorders, and substance abuse disorders [4,8]. ADHD accounted for 96,625 disability-adjusted life years (DALYs) in Europe in 2019 at a rate of 11.4 per 100,000 (0.03% of DALYs for all causes) [9].

While pharmacological treatment is the mainstay in ADHD management, nonpharmacological treatments, such as behavioral parent training and social skills training, are also used. They are most commonly recommended for very young children or for mild to moderate ADHD and can also be used in combination with medication. Cognitive behavioral therapy can be useful in adult patients; however, data suggest nonpharmacological treatments are less effective than pharmacological treatments and more research is needed. Psychostimulants (methylphenidate and amphetamines) are the most common pharmaceuticals used in the management of ADHD, followed by atomoxetine, guanfacine, and clonidine. Psychostimulant treatment is efficacious in the short term, but evidence for long-term efficacy is less clear [8].

While there is consistent evidence that ADHD has high heritability, most cases are considered to be multifactorial in origin [4]. Environmental factors, such as toxins and chemicals, have been of interest in recent research. It is hypothesized that environmental exposures, which may cause adverse outcomes, occur in utero or in early childhood and hence these time periods are the primary targets of investigation [10,11].

The European Human Biomonitoring Initiative (HBM4EU) (https://www.hbm4eu.eu/) (accessed on 13 January 2022) is a joint European program of 30 countries (26 European Member States, three associated countries, and Switzerland) and the European Environment Agency, co-funded by the European Commission and national governments [12]. The HBM4EU initiative coordinates and advances human biomonitoring in Europe to provide better evidence about the actual exposure of citizens to chemicals and the potential health effects. The initiative aims to contribute directly to the improvement of health and well-being of all citizens, by investigating how exposure to chemicals affects the health of different vulnerable groups, such as children and pregnant women, as well as of highly exposed groups such as workers.

In the framework of HBM4EU, 18 chemical substances or chemical groups have been prioritized for investigation, in order to provide new knowledge to support policy decisions at the European level. The aim of this scoping review was to summarize known or possible associations between ADHD and environmental exposures related to HBM4EU priority substances.

## 2. Materials and Methods

This scoping review was prepared in the framework of the HBM4EU initiative (https://www.hbm4eu.eu/) (accessed on 13 January 2022) [12]. In the HBM4EU project, eighteen priority substances have been selected: acrylamide, anilines, aprotic solvents, arsenic (As), benzophenones (UV filters), bisphenols, cadmium (Cd), chromium VI (Cr VI), flame retardants, lead (Pb), mercury (Hg), mycotoxins, per- and polyfluoroalkyl substances (PFASs), pesticides, phthalates and Hexamoll ^®^ DINCH, polycyclic aromatic hydrocarbons (PAHs), chemical mixtures, and emerging chemicals [13].

Literature searches were performed in PubMed in June–September 2020 and repeated in Spring 2021 to include any relevant new publications. The search used the terms “ADHD”, “meta-analysis”, or “review”, and each of the HBM4EU priority substances separately. Additional searches were conducted with more general search terms, such as “environmental chemical” and “exposure”. Only meta-analyses and review articles were included, as they provide more extensive evidence compared to individual studies. The included articles have been published since 2010. However, reviews or meta-analyses published since 2010 also include studies published earlier. Only human studies were included.

The literature searches were not conducted in a manner that fulfilled the criteria of a systematic review. This is a scoping literature review [14], which summarizes epidemiological studies on the associations of ADHD and HBM4EU priority substances, or their possible associations, which justify further research. These substances included lead, phthalates, bisphenol A (BPA), polycyclic aromatic hydrocarbons (PAHs), flame retardants, mercury, cadmium, pesticides, and per- and polyfluoroalkyl substances (PFASs). These substances were then classified into three categories, based on the appraised level of evidence: (1) moderate to high, (2) limited, or (3) low level of evidence of an association between the substance and ADHD.

## 3. Results

In this section we present the evidence of the association between selected HBM4EU priority substance chemicals and ADHD. A brief description of each chemical is provided.

The collected evidence indicates a moderate to high level of association between lead (Pb), phthalates or bisphenol A (BPA) and ADHD. There is limited evidence for an association between ADHD and PAHs, flame retardants, mercury (Hg), and pesticides. The evidence of association between ADHD and cadmium (Cd) and PFASs is low, but further research is justified.

### 3.1. Substances with Moderate to High Evidence of Association with ADHD

#### 3.1.1. Lead (Pb)

Lead is a metal element that exists in inorganic and organic forms. Although it is naturally occurring, its widespread occurrence is due to human activity, such as its historic use in lead paint and petrol. Food is the major source of exposure and there is no recommended tolerable intake level. Lead exposure can cause developmental neurotoxicity and adult nephrotoxicity [15], may damage fertility or the unborn child, causes damage to organs through prolonged or repeated exposure, may cause cancer, and may cause harm to breast-fed children [16]. Lead is a substance of very high concern (SVHC) and included in the Candidate List for Authorisation (https://www.echa.europa.eu/candidate-list-table) (accessed on 13 January 2022).

Two recent reviews have been done on the association between lead exposure and ADHD (see Table 1). A systematic review by Donzelli et al. in 2019 identified 17 studies published between 2013 and 2018 [17]. The exposure in most studies was measured during early childhood but a study assessing prenatal exposure was also included. A systematic review and meta-analysis by Nilsen and Tulve (2020) analyzed 14 studies between 1980 and 2017 [18]. Only studies assessing exposure during childhood were included. Both reviews concluded that there is evidence for an association between lead exposure and ADHD.

#### 3.1.2. Phthalates

Phthalates are a group of chemicals used as plasticizers in the production of plastics to make products more flexible and durable. Their use is widespread, and phthalates can be found in a variety of everyday products such as food containers, toys, personal care products, pharmaceutical products, and many other non-food consumer products. Phthalates can migrate into food, air, and water, and humans can be exposed through ingestion, dermal uptake and inhalation [19]. Phthalates can have a variety of adverse impacts on human health and laboratory animals [20,21], the most prominent of which are endocrine disrupting and reproductive effects [22]. Several epidemiological studies have described associations between phthalate exposure and obesity, insulin resistance, and asthma [23,24,25].

A recent systematic review investigated the epidemiological evidence of an association between prenatal and childhood phthalate exposure and ADHD in children (Table 1) [26]. While a meta-analysis was not conducted due to heterogeneity of study designs and methods, 14 out of the 16 included studies reported an association between phthalates and ADHD. Four studies reported a stronger association in boys than girls. High molecular weight phthalates such as di(2-ethylhexyl) phthalate (DEHP) were consistently associated with adverse effects.

#### 3.1.3. Bisphenol A

Bisphenol A (BPA) is used in the production of polycarbonate plastics and epoxy resins, which are used in a wide range of consumer products. The presence of BPA is ubiquitous in the environment and can enter the body through ingestion, dermal uptake and inhalation. There is evidence of multi-system toxicity of BPA in humans, with the reproductive system being a key target organ of BPA [27].

A systematic review from 2018 (see Table 1) found 3 human studies on the relationship between early-life BPA exposure and hyperactivity [28]. Each of these studies found significant effects of BPA exposure on increased hyperactivity; however, there were differences between sexes and windows of exposure.

In addition, phthalates and BPA had been analyzed in a combined category in one review and meta-analysis (Table 1) [18]. The results indicated an increase in ADHD in relation to this combined category.

**Table 1 ijerph-19-02849-t001:** Summary of studies providing moderate to high evidence of the association between ADHD and each of lead (Pb), phthalates (PhPl) and bisphenol A (BPA).

Substance	Study	Countries	*n*, Setting, Target Group	Main Findings and Conclusions
Lead (Pb)	Donzelli et al. (2019). The Association between lead and Attention-Deficit/Hyperactivity Disorder: A Systematic Review. Int J Environ Res Public Health. 2019;16:382 [17]	Mexico China USA South Korea Germany Spain Belgium Taiwan Turkey	*n* of studies: 17 *n* of participants: 8940 (range 117–2195) Setting: 2 cross-sectional, 5 birth cohort, 10 case–control Target group: exposure assessment to lead performed during pregnancy and early childhood	12 studies showed positive associations between lead exposure and ADHD whereas the remaining 5 studies found no association. The authors conclude there is an association between lead and ADHD and even low levels of lead raise the risk.
Pb	Nilsen and Tulve (2020). A systematic review and meta-analysis examining the interrelationships between chemical and non-chemical stressors and inherent characteristics in children with ADHD. Environ Res. 2020;180:108884 [18]	Canada South Korea China Mexico Romania USA India United Arab Emirates	*n* of studies: 13 on lead *n* of participants: 25,253 ^1^, 17,158 ^2^ Setting: observational Target group: infants, children, and adolescents (for lead, ages 3 to 12 included)	The overall OR for Pb exposure being associated with an ADHD diagnosis was 3.39 (90% CI 2.66–4.12, *p* < 0.001). The OR for specific ADHD diagnoses was 4.06 (2.89–5.23, *p* < 0.001). The certainty of evidence in ADHD-specific and all-symptom meta-analyses was determined to be ‘moderate’.
Phthalates	Praveena et al. (2020) Phthalates exposure and attention-deficit/hyperactivity disorder in children: a systematic review of epidemiological literature. Environ Sci Pollut Res Int. 2020;27(36):44757–44770. [26]	Canada South Korea USA Central Taiwan Japan Belgium Norway China	*n* of studies: 16 *n* of participants: 7019 (range 122–1318) Setting: cross-sectional, cohort, case–control Target group: infants, children, adolescents (5 weeks to 16 years)	14 studies found an association between phthalate exposure and ADHD. One study found no significant association between phthalate concentration and ADHD at age five. The remaining study found sex-specific associations with behavior scores among children. The authors conclude that despite the observed associations, due to study limitations, it is difficult to produce definite conclusions.
Bisphenol A (BPA)	Rochester et al. (2018). Prenatal exposure to bisphenol A and hyperactivity in children: a systematic review and meta-analysis. Environ Int. 2018;114:343–356 [28]	Spain USA	*n* of studies: 3 human studies *n* of participants: 1151 (range 237–657) Setting: prospective/longitudinal birth cohort Target group: exposure during gestation or childhood	The systematic review among human studies found that early BPA exposure is associated with a presumed hazard of hyperactivity in humans. All three human studies showed significant effects of BPA on hyperactivity.
Phthalates/BPA ^3^	Nilsen and Tulve (2020). A systematic review and meta-analysis examining the interrelationships between chemical and non-chemical stressors and inherent characteristics in children with ADHD. Environ Res. 2020;180:108884 [18]	USA (two studies on BPA) China (BPA) China (Ph) South Korea (Ph)	*n* of studies: 5 studies on phthalates/plasticizers (bisphenol A also included in this category) *n* of participants: 21,594 Setting: observational Target group: infants, children and adolescents (ages 6 to 18), only studies examining exposure after birth included	OR for ADHD diagnosis associated with phthalates/plasticizers in both sexes was 3.31 (2.59–4.02, *p* < 0.0001). The level of certainty was ‘moderate’, and these results indicate an increase in ADHD with phthalates/plasticizers exposure.

^1^ DSM ADHD (total), hyperactivity, impulsiveness and inattention diagnoses (DSM-IV or comparable system used). ^2^ DSM ADHD (total) diagnoses (DSM-IV or comparable system used). ^3^ This study analyzed studies on phthalates and bisphenol A in a combined “phthalates/plasticizers” category; odds ratio, OR; confidence interval, CI; DSM, Diagnostic and Statistical Manual of Mental Disorders.

### 3.2. Substances with Limited Evidence of Association with ADHD

#### 3.2.1. Polycyclic Aromatic Hydrocarbons (PAHs)

Polycyclic aromatic hydrocarbons (PAHs) are a class of organic compounds that are released into the air during incomplete combustion of organic matter, such as coal, fuel, tobacco and meat. The PAH class is considered carcinogenic and multitoxic to humans [29]. Primary routes of exposure are through inhalation and ingestion [30].

Two systematic reviews showed limited evidence of an association of PAHs and ADHD (Table 2) [31,32]. Both reviews included studies on both prenatal and postnatal PAH exposure. One of the reviews also included meta-analysis, but the results from an analysis combining five studies did not quite reach statistical significance [31].

#### 3.2.2. Flame Retardants

Flame retardants (FR) are chemical compounds that are added to consumer products or materials to reduce flammability and are commonly used in e.g., building materials, textiles, electronics, vehicles, insulation, and appliances. Human exposure occurs via inhalation, ingestion, and dermal exposure.

Among the most common FRs since the 1970s were polybrominated diphenyl ethers (PBDEs). PBDEs have been identified as potential causes of neurotoxic, carcinogenic, and endocrine effects. Due to safety concerns surrounding toxicity, persistence, and bioaccumulation, PBDEs are now restricted and being phased out of use. Organophosphate esters (OPE) are now more commonly used, and more data on human toxicity are needed [38,39].

FRs is a large group of chemicals with a common function but not necessarily common chemical structure. Only sub-categories of FRs have been investigated for their association with ADHD and other cognitive/behavioral conditions, captured in two systematic reviews investigating flame retardants and attention problems (Table 2). Lam et al. (2017) reviewed 9 studies, conducted in the USA, Spain, and the Netherlands, that assessed prenatal and postnatal exposure to polybrominated diphenyl ethers (PBDEs) and their association with ADHD, intelligence and attention-related behavioral conditions [33]. The results of the studies were inconsistent. The overall strength of the evidence was considered “limited” for the outcome of ADHD and attention-related behaviors. The possibility of publication bias could not be ruled out, but the quality of the studies included was moderate.

Doherty (2019) assessed the link between children’s health and organophosphate esters (OPEs) before birth and in childhood in a review of two studies from the USA [34]. While the conclusion was that there was an association between greater hyperactivity and organophosphate ester metabolites, the authors noted that further research is required to make conclusions about the potential cognitive and behavioral effects of early life OPE exposure.

#### 3.2.3. Mercury

Mercury (Hg) is a heavy metal element which occurs naturally but is also spread as a pollutant as a result of human activity. Coal-fired powerplants, waste incinerators, and small-scale gold mines are significant sources of mercury in the environment [40]. Methylmercury (MeHg) readily bioaccumulates in the food chain through plants and aquatic ecosystems and the main route of human exposure is through ingestion from contaminated seafood [41,42].

Mercury is neurotoxic, causing a variety of neurological symptoms depending on the level and route of exposure. Children are more susceptible than adults, and transplacental exposure in utero is particularly dangerous. The World Health Organization classifies mercury exposure as a major public health concern [40].

Significant associations between mercury exposure and ADHD were observed in two meta-analyses (Table 2) [18,35]. In one meta-analysis, studies where the exposure was assessed during the prenatal period were excluded [18], while the other meta-analysis included both prenatal and postnatal exposure assessments [35]. These meta-analyses only included two or three studies, and therefore the results should be interpreted with caution.

#### 3.2.4. Pesticides

Pesticides are a very large group of chemicals used to control unwanted weeds, insects, various pests and disease carriers, such as rats and mosquitoes [43,44]. Agriculture is the largest consumer of pesticides [44]. The general population is exposed to low amounts by consuming pesticide residues in food, and people working in agriculture may have high dermal and inhalation exposure. Developmental neurotoxicity, carcinogenicity, and endocrine disruptive properties of pesticides are of particular concern [45]. Studies have found associations between pesticides and certain cancers such as bladder cancer and acute leukemia, exacerbations of asthma, type 2 diabetes, and neurological conditions such as Parkinson’s disease [44].

Based on results from a review and meta-analysis including six studies where pesticide exposure was assessed during childhood [18] and another review with 12 studies with studies on both prenatal and postnatal pesticide exposure [36] the evidence for an association between the exposure to pesticides and ADHD was low (Table 2). However, the evidence from a review with 15 studies on ADHD suggested a possible link between pesticides and ADHD [37]. More specifically, the included studies suggested that exposure to a number of (but not all) pesticides in different periods of pregnancy or in infancy/childhood may affect behavior and neurodevelopment in childhood.

### 3.3. Substances with Low Evidence of Association with ADHD

#### Cadmium and Per- and Polyfluoroalkyl Substances 

The evidence of the association between ADHD and cadmium (Cd) and per- and polyfluoroalkyl substances (PFASs) was low based on studies identified from the literature review.

In a review of studies on the association of perinatal and childhood exposure to cadmium and ADHD or ADHD-like behavior, none of the included five studies reported a significant association [46].

Two studies on perfluoroalkyl compounds (PFCs) (i.e., PFASs) during childhood and ADHD were included in a review by Nilsen and Tulve (2020) [18]. The results were somewhat discordant: One study reported no significant relationship between increasing PFC exposures and ADHD outcomes in children aged 6–12 years but observed a sex-based relationship. In the other study, a significant relationship in children aged 12–15 years was observed, whereas sex-based differences were not discussed. Five studies in which exposure was assessed either before birth or during childhood were identified in another review with a broader outcome definition [47]. Several, but not all, studies suggested a positive association between prenatal exposure to PFCs and ADHD or related behavioral problems.

## 4. Discussion

This scoping review demonstrates that there are several chemicals, which children and adults encounter in their living and working environment that may be associated with ADHD outcomes. For certain environmental pollutants, such as lead, phthalates, and bisphenol A, the association is more robust and the evidence stronger. For several other chemicals, the evidence is weaker but warrants more research.

The starting point for the current review was the list of priority chemicals of the HBM4EU project (https://www.hbm4eu.eu/) (accessed on 13 January 2022) [12]. Reviews and meta-analyses on the association of these chemicals with ADHD were compiled in the current paper. We aimed to describe the conclusions drawn in the published review and meta-analysis articles, and therefore refrained from interpreting the results from the original individual studies included in the reviews.

The inclusion criteria and other methods of the included reviews varied. Furthermore, many reviews highlighted the heterogeneity of the individual studies. The quality of the studies included in the reviews was also variable. Sample sizes varied from a few dozen to several thousand and many different study designs were included. Different studies and reviews also used different definitions of ADHD and different methods of diagnosis or measures of outcome, which hindered forming a final conclusion based on their findings. Furthermore, adjusting for confounding factors varied between the original studies and was not consistently reported in all included reviews.

The inclusion of published reviews in the current paper was not straightforward in all cases. The categorization of chemicals varied between the reviews. As an example, studies on phthalates and BPA were included in Nilsen and Tulve’s 2020 systematic review and meta-analysis (*n* = 5); however, they were analyzed together under the name ‘phthalates/plasticizers’ [18], although BPA is considered neither a plasticizer nor a phthalate [48]. An OR of 3.31 (2.59–4.02) was found for ADHD diagnosis associated with this category.

As some of the included systematic reviews and meta-analyses were performed on very similar research questions and similar inclusion criteria, there is some overlap in the studies included. Thereby, the evidence may not be as extensive as the number of studies reported for each included review might suggest. There was some overlap for lead, significant overlap for pesticides and marked overlap for PAHs (for details, see Appendix A).

Individual studies with notable results that have not yet been included in systematic reviews, have been published since the publication of the reviews that were included in the current paper. In a Norwegian cohort study on pesticides, Choi et al. (2021) demonstrated that children born to mothers who had elevated urinary DPHP (diphenyl phosphate, an OPE metabolite) in mid-pregnancy had a higher risk of ADHD [10]. The findings in this study were also in line with the hypothesis that girls are more vulnerable to ADHD from DPHP exposure. This new study strengthens the suggestive conclusion of a review included in the current paper on possible effects of pesticide exposure during pregnancy on neurodevelopment in childhood [37].

For cadmium, none of the studies included in a systematic review that we identified reported a significant association between perinatal and childhood exposure to cadmium and ADHD or ADHD-like behavior [46]. However, a significant association was observed in two individual studies, which were published after the review [49,50]. Further studies are needed to confirm the association.

It was difficult to make precise conclusions in terms of the strength of evidence of association between certain chemicals and ADHD. It was particularly challenging to distinguish between ‘limited’ and ‘low’ evidence. It is apparent that the chemicals in these categories require more research input. An issue worth mentioning here is that the timing of chemical exposure, e.g., prenatal or during childhood, may be especially important with regards to ADHD development. Thereby, if the exposure was measured during a less relevant time period in a given study, the association may have appeared weaker than it actually is.

While not the focus of this paper, the mechanisms by which chemicals may affect the development of ADHD are also of great interest. How much is known about the mechanisms varies depending on the chemical, and the mechanisms and chemical pathways also vary. For example, serotonergic and dopaminergic systems are possible targets through which BPA may exert its neurological effects in humans [28]. Pesticides may also target neurotransmitters such as dopamine, noradrenaline, and serotonin, and the role of the cholinergic system has also been studied [37]. Many metals, e.g., Pb, Hg, Cd, have been implicated, and they may have a common mechanism of action in relation to ADHD, possibly via interfering with calcium channels in neurotransmitter regulation and the N-methyl-D-aspartate receptor (NMDAR) [18].

The focus of this scoping review was only on those selected as the priority substances of the HBM4EU project. Therefore, many substances for which at least suggestive associations with ADHD have been seen, were excluded from the current paper, such as paracetamol, parabens, and dioxin, as well as smoking during pregnancy [51,52,53,54,55]. Only those HBM4EU priority substances were included in the current paper for which relevant reviews or meta-analyses on their associations with ADHD were found. For some of the HBM4EU priority substances the research literature was scarce at the time of the literature review. Therefore, associations may exist between ADHD and other chemicals which have not yet been investigated or at least not described in review articles. It should be highlighted that the aim of this paper was to compile and describe the conclusions of previous reviews and meta-analyses but not to evaluate the quality of the evidence. As the effects of chemicals are constantly examined and evidence of their possible associations with health outcomes is accumulating, more reliable conclusions may be possible in future.

Some evidence came from studies designed for other research purposes than assessing possible associations of a particular chemical and ADHD. In future, it would be valuable to implement hypothesis-driven, properly designed studies to collect stronger evidence of associations or the lack of an association between chemical exposures and ADHD.

Furthermore, more research is needed in Europe to enhance the understanding of the situation more locally. So far, a large part of the research has been done in North America. Currently, the HBM4EU initiative strives to fill this gap by collecting information on the exposure of European populations to the priority substances and their health effects to support policy decisions for public health protection.

## 5. Conclusions

Thus far, evidence from epidemiological studies suggests there is an association between ADHD and a number of chemicals, with moderate to high evidence for at least lead, phthalates and bisphenol A.

The methods of the individual studies included in the reviews and meta-analyses covered in the current paper varied considerably, which complicated drawing final conclusions.

More research is needed for many chemicals to collect stronger evidence of associations or the lack of an association between chemical exposures and ADHD.

More European studies are needed. The HBM4EU has prepared standardized operating procedures for collection of data on chemical exposure on humans, and also provides guidelines for standardized collection of health outcomes. The project generates new evidence of chemical exposure and health effects in European countries.

## Figures and Tables

**Table 2 ijerph-19-02849-t002:** Summary of studies providing evidence of limited association between ADHD and polycyclic aromatic hydrocarbons (PAH), flame retardants (FR), mercury (Hg), and pesticides.

Substance	Study	Countries	*n*, Setting, Target Group	Main Findings and Conclusions
Polycyclic aromatic hydrocarbons (PAH)	Rezaei Kalantary et al. (2020). Association between exposure to polycyclic aromatic hydrocarbons and attention deficit hyperactivity disorder in children: a systematic review and meta-analysis. Environ Sci Pollut Res Int. 2020;27:11531–11540 [31]	Spain USA	*n* of studies: 6 in the systematic review and 5 in the meta-analysis *n* of participants: 2799 (range 242–1257) Setting: cohort and cross-sectional studies Target group: children	There was no significant association between PAH exposure and ADHD for children when all studies were combined (overall odds ratio = 1.99, 95% CI = 0.96–4.11) with low heterogeneity (I2 = 28.73%; *p* < 0.001). However, the authors concluded that PAH have a vital role in child ADHD.
PAH	Aghaei et al. (2019). Association between ambient gaseous and particulate air pollutants and attention deficit hyperactivity disorder (ADHD) in children; a systematic review. Environ Res. 2019;173:135–156 [32]	Spain USA	*n* of studies: 7 *n* of participants: 2944 (range 215–1257) Setting: cohort and cross-sectional studies Target group: children	Eleven associations were investigated in seven studies related to PAH in which five significant associations were observed. On the other hand, no associations were found in six investigations. Although limited evidence of detrimental effects of PAH on ADHD was found in this review, moderate-quality studies in terms of exposure assessment showed positive associations.
Flame retardants (FR)	Lam et al. (2017). Developmental PBDE Exposure and IQ/ADHD in Childhood: A Systematic Review and Meta-analysis. Environ Health Perspect. 2017;125(8):086001 [33].	USA Spain Netherlands	*n* of studies: 9 *n* of participants: 2033 ^1^ (range 43–622) Setting: cohort and cross-sectional studies Target group: mother–child pairs	In this systematic review developmental exposure to polybrominated diphenyl ethers (PBDEs) and intelligence or ADHD and attention-related behavioral conditions in humans were reviewed. The body of evidence was of moderate quality for ADHD with “limited” evidence for an association with PBDEs.
FR	Doherty (2019). Organophosphate Esters: Are These Flame Retardants and Plasticizers Affecting Children’s Health? Curr Environ Health Rep. 2019;6:201–213 [34]	USA	*n* of studies: 2 *n* of participants: 488 ^1^ (range 199–282) Setting: prospective pregnancy cohorts Target group: children	Greater hyperactivity was associated with organophosphate ester metabolites. However, the available studies have limitations. More evidence needed.
Mercury (Hg)	Yoshimasu et al. (2014). A meta-analysis of the evidence on the impact of prenatal and early infancy exposures to mercury on autism and attention deficit/hyperactivity disorder in the childhood. Neurotoxicology. 2014;44:121–131 [35].	Canada USA	*n* of studies: 2 studies with 3 data sets for methylmercury with relevant exposure periods *n* of participants: 1067 (range 279–788) Setting: cohort study Target group: children	Environmental methylmercury exposure showed moderate adverse effects on ADHD (OR 1.60, 95% CI 1.10–2.33). With a low number of studies, the authors advise cautious interpretation of results.
Hg	Nilsen and Tulve (2020). A systematic review and meta-analysis examining the interrelationships between chemical and non-chemical stressors and inherent characteristics in children with ADHD. Environ Res. 2020;180:108884 [18]	Canada South Korea United Arab Emirates	*n* of studies: 3 studies on mercury *n* of participants: 2139 Setting: observational Target group: age range 5–15 years	The OR for Hg exposure and all ADHD outcomes was 2.68 (2.16–3.19, *p* < 0.0001). The authors conclude that despite significant results, the number of studies is small, and a relationship cannot be proven.
Pesticides	Nilsen and Tulve (2020). A systematic review and meta-analysis examining the interrelationships between chemical and non-chemical stressors and inherent characteristics in children with ADHD. Environ Res. 2020;180:108884 [18]	USATaiwan	*n* of studies: 6 *n* of participants: 6440 (range 117–2546) Setting: cross-sectional and case–control studies Target group: children aged 4–17 years	Pesticides were included in the group of organic contaminants (OCs). Practically no effect for OCs on ADHD outcome was ascertained (OR 0.99, 90% CI 0.96–1.02). The certainty of the evidence was considered ‘very low’ by GRADE standards.
Pesticides	Roberts, J.R. et al. (2019). Children’s low-level pesticide exposure and associations with autism and ADHD: A review. Pediatric Research, 85(2), 234–241 [36].	USA Costa Rica Taiwan Spain Belgium	*n* of studies: 12 *n* of participants: 13,007 (range 140–5193) Setting: cross-sectional, case–control and cohort studies Target group: children and mother–child pairs	In most of the included studies the outcome was ADHD, but also hand tremor, as an example, was used as a sign of ADHD. The level of pesticide exposure in the studies was generally low. Some kind of association between pesticides and ADHD or measures of attention was found in the majority of studies. However, the evidence could not be considered conclusive.
Pesticides	Tessari et al. Association Between Exposure to Pesticides and ADHD or Autism Spectrum Disorder: A Systematic Review of the Literature. J Atten Disord. 2022 [37].	Taiwan USA Canada Costa Rica Mexico Belgium Greenland Poland Ukraine France Slovakia Spain Norway Germany	*n* of studies: 15 *n* of participants: 15,132 (range 117–4437) Setting: cross-sectional, case–control and cohort studies Target group: children and mother–child pairs	Ten out of 15 studies reported a significant association between exposure to pesticides and ADHD/ADHD symptoms. The strengths of the association and the possible confounders controlled for varied substantially across studies. The evidence suggested a possible link between pesticides and ADHD, but heterogeneity across studies prevented firm conclusions.

^1^ Where a range of participants was reported depending on exposure matrix or outcome, the higher number was used for the calculation.

## Data Availability

Not applicable.

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
