# Peer review of "The Association between ADHD and Environmental Chemicals—A Scoping Review"

_ijerph, 2022, doi:10.3390/ijerph19052849_

Round 1

Reviewer 1 Report

This scoping review summarized the results of 17 meta-analyses and review studies on the associations between ADHD and environmental exposures to substances selected as priority chemicals of the European Human Biomonitoring Initiative (HBM4EU). This scoping review had the conclusions consistent with the evidence and arguments presented with references appropriate references.

Because of focusing on the meta-analyses and review studies, this scoping review provides more extensive evidence compared to those only examining individual studies.

Moreover, this scoping review had a clear target by focusing on the substances selected as priority chemicals of the HBM4EU.

I would like to suggest the authors to add more introductions about the possible pathways for the influences of exposures to chemical substances on the development of ADHD. Although it is not the main aim of this study, adding the introduction will help the readers to know the importance of this study.

Author Response

Response:

We thank the reviewer for the positive comments. We also think that the possible pathways for the influences of exposures are essential. However, in this review, our focus was to review the literature of studies on only the associations of selected substances and ADHD. The possible pathways also vary between the chemicals. We wanted to keep the introduction fairly compact but we have now added discussion on this relevant matter as follows:

“While not the focus of this paper, the mechanisms by which chemicals may affect the development of ADHD are also of great interest. How much is known about the mechanisms varies depending on the chemical. For example, serotonergic and dopaminergic systems are possible targets through which BPA may exert its neurological effects in humans (Rochester, 2018). Pesticides may also target neurotransmitters such as dopamine, noradrenaline and serotonin, and the role of the cholinergic system has also been studied (Tessari, 2020) Many metals, e.g. Pb, Hg, Cd, have been implicated, and they may have a common mechanism of action in relation to ADHD, possibly via interfering with calcium channels in neurotransmitter regulation and the N-methyl-D-aspartate receptor (NMDAR) (Nilsen & Tulve, 2020).”

Reviewer 2 Report

 Manuscript : The association Between ADHD and Environmental Chemicals by Moore e.a.
Review:

Very nice review of the literature about the association between ADHD en Environmental Chemicals focusing on the priority chemicals of the HBM4 EU study.

In the introduction I miss the relation with smoking in pregnancy and later ADHD in children, a well established finding. And nicotine is blamed. Thinking of this finding I wonder what the effects are of the neonicotinoids. This is not separately addressed in this review when the pesticides are discussed?? I think this is an aspect that must be addressed.

Another (in your eyes old-fashioned relation) is the use of paracetamol in pregnancy and ADHD , so well demonstrated by Jean Golding in the ALSPAC- study , maybe this finding can be included in the introduction.

Otherwise the article is fine and very well written.

References:
Heavy nicotine exposure (cotinine level > 50 ng/mL) was associated with the highest risk of offspring ADHD, with an odds ratio of 2.21 (95% CI 1.63-2.99) in the adjusted analyses.18 jun. 2019

Smoking During Pregnancy and Risk for ADHD in Children

https://womensmentalhea

And: Paper by Jean Golding about pracetamol

'Associations between paracetamol (acetaminophen) intake between 18 and 32 weeks gestation: a longitundinal cohort studyby Jean Golding, Steven Gregory, Rosie Clark et al published in Paediatric and Perinatal Epidemiology 

Fine article, but I miss two in my eyes important aspects , first in the chapter on pesticides separately must be addressed the neonicotinoids and second, in the introduction both the negative effects resulting from smoking in pregnancy  and the use of paracetamol must be addressed.

Last point: a less important contribution on perinatal dioxin exposure and ADHD: 

Koppe JG, Leijs M, ten Tusscher GW, Olie K, van Aalderen WMC, de Voogt P,           Tom     Vulsma T, Legler J ,  Van de Bor M, Van der Ven L, Koppe JG

            Perinatal dioxin exposure in the Netherlands and attention deficit hyperactivity   disorder in the offspring. Organohalogen Compounds 2012;74:1452-1455.

Author Response

Response:

We thank the reviewer for relevant comments. However, our focus was only on chemicals and chemical groups selected as priority chemicals of the European Human Biomonitoring Initiative (HBM4EU). We are aware that the priority chemicals of HBM4EU do not cover all chemicals which have adverse health effects. However, we think it was necessary to outline the topic of our review by clearly defining the included chemical groups. As some of the chemicals suggested by the reviewer are also essential in regards to ADHD, we added discussion on them:

“Therefore, many substances for which at least suggestive associations with ADHD have been seen, were excluded from the current paper, such as paracetamol, parabens and dioxin as well as smoking during pregnancy (Golding et al 2020, Baker et al 2020, Huang et al 2018, Sourander et al 2019).”

Regarding neonicotinoids, in our review, pesticides are covered as a group of chemicals to be in line with the included reviews. Thus, we did not deal with each chemical used as a pesticide separately as the previously published reviews covered them as a group of chemicals.

Reviewer 3 Report

This review provides significant information on the effects of environmental toxicants on child development. Although the review complied with all the literature that reported necessary information on the common environmental toxicants, it lacks data on some other CDC-listed environmental chemicals, including N,N-Diethyl-meta-toluamide (DEET), Triclosan, 4-tert-Octylphenol, Herbicides, Parabens, Perchlorate, and Volatile Organic Compounds that have health consequences in child development. 

Author Response

Response

We thank the reviewer for this comment. We are aware that our scoping review does not cover all relevant chemicals. However, our focus was only on chemicals and chemical groups selected as priority chemicals of the European Human Biomonitoring Initiative (HBM4EU). We also think it was necessary to outline the topic of our review by clearly defining the included chemical groups.

Regarding DEET, in our review, pesticides are covered as a group of chemicals to be in line with the included reviews. Thus, we did not deal with each chemical used as a pesticide separately. Furthermore, herbicides (“weedkillers”) are included in the group of pesticides as described in the beginning of chapter 3.2.4 Pesticides: “Pesticides are a very large group of chemicals used to control unwanted weeds, insects, various pests and disease carriers, such as rats and mosquitoes[43,44].”

We made literature searches on other suggested chemicals, but did not find many studies among humans showing significant associations with ADHD. For parabens, we found an interesting study by Baker et al (2020). We added discussion on parabens and some other relevant chemicals, which are not included among the HBM4EU priority substances as follows:

“Therefore, many substances for which at least suggestive associations with ADHD have been seen, were excluded from the current paper, such as paracetamol, parabens and dioxin as well as smoking during pregnancy (Golding et al 2020, Baker et al 2020, Huang et al 2018, Sourander et al 2019).”